# Double-Bilayer polar nanoregions and Mn antisites in $(Ca, Sr)_3Mn_2O_7$

Leixin Miao [1], Kishwar-E Hasin[2], Parivash Moradifar[1,3], Debangshu Mukherjee [4], Ke Wang[5], Sang-Wook Cheong [6], Elizabeth A. Nowadnick[2] & Nasim Alem [1,5] ✉

The layered perovskite $Ca_3Mn_2O_7$ (CMO) is a hybrid improper ferroelectric candidate proposed for room temperature multiferroicity, which also displays negative thermal expansion behavior due to a competition between coexisting polar and nonpolar phases. However, little is known about the atomic-scale structure of the polar/nonpolar phase coexistence or the underlying physics of its formation and transition. In this work, we report the direct observation of double bilayer polar nanoregions (db-PNRs) in $Ca_{2.9}Sr_{0.1}Mn_2O_7$ using aberration-corrected scanning transmission electron microscopy (S/TEM). In-situ TEM heating experiments show that the db-PNRs can exist up to 650 °C. Electron energy loss spectroscopy (EELS) studies coupled with first-principles calculations demonstrate that the stabilization mechanism of the db-PNRs is directly related to an Mn oxidation state change (from 4+ to 2+), which is linked to the presence of Mn antisite defects. These findings open the door to manipulating phase coexistence and achieving exotic properties in hybrid improper ferroelectric.

The development of multiferroic materials aims to realize direct electric-field controlled switching of magnetization at room temperature[1,2]. However, identifying stable and single-phase multiferroic materials to achieve this objective has been challenging because electrical polarization and magnetization must be strongly coupled together in such systems[1–6]. Recently, a new class of materials, called hybrid improper ferroelectrics, was theoretically proposed as a potential room-temperature multiferroic system[2]. The principle of the hybrid improper ferroelectric mechanism is to induce ferroelectricity, ferromagnetism, and magnetoelectricity simultaneously with the same set of lattice instabilities[2,7]. In particular, oxygen octahedron rotations and tilts, which are ubiquitous in perovskite materials and couple strongly to magnetism, combine with a layered crystal structure to induce polarization[8,9]. Hybrid improper ferroelectricity at room temperature was first experimentally confirmed in the $n = 2$ Ruddlesden-

Popper material $Ca_3Ti_2O_7$[10] and was subsequently observed in $Sr_3Sn_2O_7$ and other layered perovskites[11–13].

$Ca_3Mn_2O_7$ (CMO) is another hybrid improper ferroelectric candidate, which was the first theoretically proposed system for achieving room temperature multiferroicity[2]. $Ca_3Mn_2O_7$ is an $n = 2$ member of the Ruddlesden Popper layered perovskite family with the general formula $(ABO_3)_n(AO)$ or $A_{n+1}B_nO_{3n+1}$. The CMO crystal structure consists of double $CaMnO_3$ perovskite blocks stacked along the [001] direction, with an extra CaO rocksalt sheet inserted after each double perovskite block. At room temperature, CMO crystallizes in the polar $A2_1am$ space group. The condensation of an out-of-phase ($a^-a^-c^0$ in Glazer notation[14]) octahedral tilting distortion (Fig. 1a left) and an in-phase ($a^0a^0c^+$) octahedral rotation distortion (Fig. 1a right) establish the $A2_1am$ symmetry and induce a polarization[2]. Figure 1b shows schematic views of the $A2_1am$ crystal structure viewed along the a

[1]Department of Materials Science and Engineering, The Pennsylvania State University, University Park, PA 16802, USA. [2]Department of Materials Science and Engineering, University of California, Merced, CA 95343, USA. [3]Department of Materials Science and Engineering, Stanford University, Stanford, CA 94305, USA. [4]Computational Sciences & Engineering Division, Oak Ridge National Laboratory, Oak Ridge, TN 37830, USA. [5]Materials Research Institute, The Pennsylvania State University, University Park, PA 16802, USA. [6]Rutgers Center for Emergent Materials and Department of Physics and Astronomy, Rutgers University, Piscataway, NJ 08854, USA. ✉e-mail: nua10@psu.edu

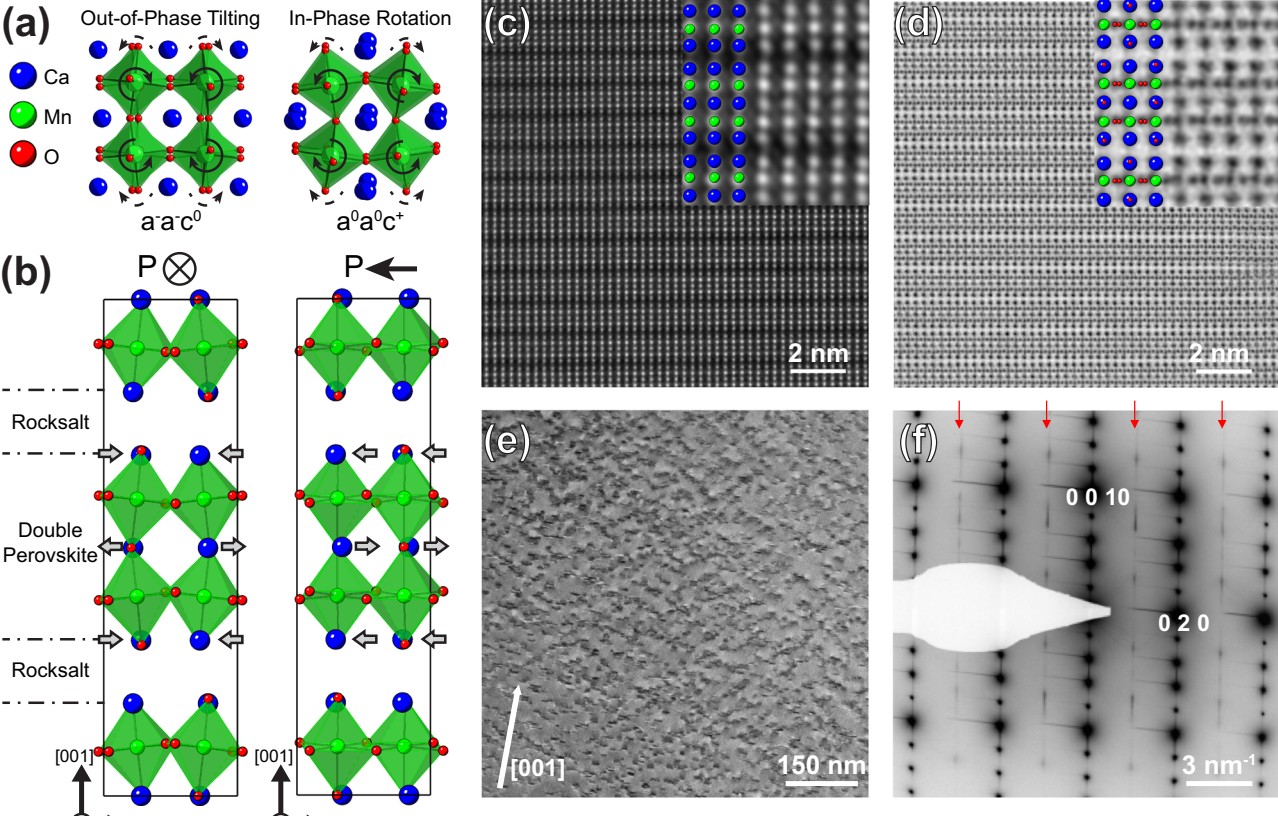

**Fig. 1 | Crystal structure illustration and structural analysis. a** Schematics of the octahedral tilting (a⁻a⁻c⁰, Glazer notation) and octahedral rotation distortions (a⁰a⁰c⁺) that establish the $A2_1am$ symmetry and induce a polarization. The solid arrows indicate the tilting/rotation of the first layer of octahedra, and the dashed arrows indicate the distortion of the successive octahedra along the viewing axis. **b** Schematics of the $A2_1am$ crystal structure for hybrid improper ferroelectric $Ca_3Mn_2O_7$ viewed along the [100] and [010] projections. The gray arrows indicate the alternating left/right and parallel displacements of the Ca atoms in each layer along [010] and [100], respectively. The alternating displacements along [010] arise from the a⁻a⁻c⁰ octahedral tilting distortion, whereas the parallel displacements

along [100] give rise to the polarization. **c, d** The ADF-STEM and ABF-STEM images of $Ca_{2.9}Sr_{0.1}Mn_2O_7$ in the $Acaa$ non-polar space group, along the [100] zone axis. The insets show the magnified micrographs with the crystal structure overlaid. **e** A TEM image of the $Ca_{2.9}Sr_{0.1}Mn_2O_7$ sample along the [100] zone axis. Numerous and randomly distributed nanometer-sized features are observed. **f** The selected area electron diffraction (SAED) pattern along the [100] zone axis. The contrast of the SAED pattern is reversed to better visualize the weak extra diffraction spots. The red arrows point out the rows of weak spots at (0, l, l), with l = 2n + 1, connected with near-continuous streaks.

([100]$_{ortho}$) and b ([010]$_{ortho}$) projections, with the polarization (P) pointing along [$\bar{1}$00]. The polarization arises primarily from a two-against-one displacement of the Ca ions along the a-axis in each perovskite block (Fig. 1b, right). In addition, the a⁻a⁻c⁰ octahedral tilting distortion involves alternating left/right displacements along the b-axis of the Ca ions in each layer (Fig. 1b, left).

Several studies have revealed the complex domain structure and phase transition sequence of CMO. Early transmission electron microscopy (TEM) studies showed twin variants of the $A2_1am$ phase with an (001) interface[15]. In addition, Gao et al. observed irregular orthorhombic twins with curved boundaries and further demonstrated the rearrangement of the oxygen octahedral tilts and rotations during a complex phase transition from $I4/mmm$, to an intermediate nonpolar phase $Acaa$ (a⁰a⁰c⁻), and finally to the polar $A2_1am$ (a⁻a⁻c⁺) phase[7]. This transition leads to a degeneracy of polarization orientations along either the a-axis ([100]$_{orth}$) or b-axis ([010]$_{orth}$) of the CMO crystal and a large number of 90° ferroelectric domain walls perpendicular to the c-axis[7]. Liu et al. experimentally demonstrated ferroelectric switching by measuring the ferroelectric hysteresis loop, but this could only be accomplished at a very low temperature (T < 28 K) due to the relatively high electrical conductivity of CMO[16]. On the other hand, the complex domain morphology of CMO leads to many novel physical properties[16,17]. Most notably, negative thermal expansion behavior was observed in both

CMO and $Ca_{3-x}Sr_xMn_2O_7$ (CSMO) crystals and was closely related to the coexistence of competing polar $A2_1am$ and nonpolar $Acaa$ phases over a large temperature range[18,19]. It was further demonstrated that negative thermal expansion could be tuned by controlling the phase competition between the polar $A2_1am$ and nonpolar $Acaa$ phase with different levels of Sr doping[19]. Although several studies have focused on understanding the macroscale properties of CMO, little has been done to uncover the structure of the polar-nonpolar phase coexistence at the atomic scale.

In this work, we uncover polar nanoregions (PNRs) in a nonpolar matrix of a layered perovskite at the atomic scale using scanning/transmission electron microscopy (S/TEM) imaging and quantify its structure with picometer precision. We further explore the underlying physics and chemistry of the phase competition transition dynamics as a function of temperature using in-situ high-resolution TEM and monochromated electron energy loss spectroscopy (EELS) techniques. We observe that the formation of Mn antisites drives the phase competition, which is a mechanism that may stabilize similar polar/nonpolar phase competition in other layered perovskite crystals and beyond.

## Results

This study explores Sr-doped CMO because the presence of the Sr cation promotes the coexistence of the $A2_1am$ and $Acaa$ phases at

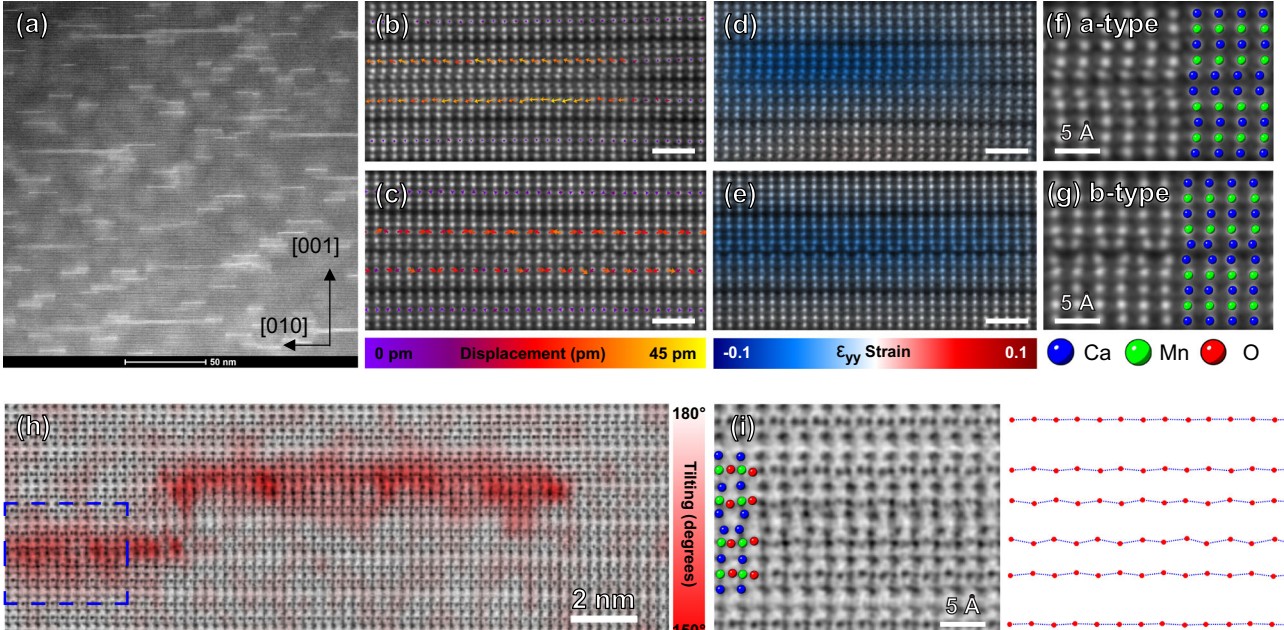

**Fig. 2 | High resolution STEM imaging and analysis of the structural distortions. a** The ADF-STEM image at lower magnification shows numerous linear structures at the [001] interface (bright lines) embedded and scattered randomly in the crystal. **b, c** The displacement measurement of the center Ca atom in the double perovskite blocks in a- and b-type db-PNRs in CSMO, respectively. The measurement is presented as colored vector maps superimposed on the ADF-STEM images of the db-PNRs, with the purple and yellow colors indicating displacements of 0 and 45 pm, respectively. The white scale bar is 1 nm. **d, e** The geometric phase analysis (GPA) $\varepsilon_{yy}$ strain mapping of the db-PNR. The white scale bar is 1 nm. **f, g** The schematics of the a- and b-type db-PNRs. **h** ABF-STEM image of the db-PNR in CSMO, with the overlaid color map on the micrograph indicating the measured oxygen octahedra tilting angles. **i** Enlarged image from the blue box in **d** highlighting the oxygen octahedra tilting at the db-PNR. The line plot with zig-zag shape on the right side is extracted from the enlarged ABF-STEM image and shows the opposite tilting directions for any two adjacent octahedra.

room temperature. The addition of 3% Sr is expected to decrease the ferroelectric transition temperature of $Ca_{2.9}Sr_{0.1}Mn_2O_7$ to below room temperature, and thus at room temperature, we expect a small fraction of the $A2_1am$ polar phase region to be embedded in a nonpolar matrix with $Acaa$ symmetry, as indicated by previous X-ray measurements[19].

Figure 1c, d show the annular dark-field (ADF) and annular bright-field (ABF) STEM micrographs of the CSMO crystal with the nonpolar $Acaa$ space group, respectively. The ADF-STEM shows the contrast from heavier atomic species, Ca, Sr, and Mn. The presence or absence of Ca/Sr displacements can be used to distinguish between the $A2_1am$ phase, where displacements are expected (Fig. 1a), and the $Acaa$ phase, where such displacements are not allowed by symmetry. The Ca/Sr atoms in the image do not exhibit displacement with regard to the unit cell center, which is expected since the matrix should belong to the nonpolar $Acaa$ space group. On the other hand, we observe many nanometer-sized features along the (001) interface distributed randomly all across the sample on the TEM micrograph taken from the $[100]_{orth}$ zone axis (Fig. 1e). To determine the crystal structure, we obtain the selected area electron diffraction (SAED) patterns from the same sample (Fig. 1f). A row of weak spots at $(0, l, l)$ (with $l = 2n + 1$) connected with near-continuous streaks is observed in the SAED patterns, suggesting the formation of a superstructure along the (001) interfacial planes, similar to the features in the electron diffraction study reported in an undoped $Ca_3Mn_2O_7$ crystal with a polar $A2_1am$ space group[15]. High-resolution transmission electron microscopy images taken in this region confirm that the weak spots and the streaks in the SAED pattern arise from the linear features within the crystalline matrix (Supplementary Fig. 1).

To directly observe the linear features at the atomic scale, we perform aberration-corrected high-resolution scanning transmission electron microscopy (AC-STEM). Numerous linear features (bright lines) along the (001) interface are observed in the ADF-STEM image in Fig. 2a. The high-resolution ADF-STEM image of the linear features reveals two distinct types of double bilayer polar nanoregions (a-type and b-type db-PNRs), as shown in Fig. 2b, c, respectively. Both types of db-PNRs consist of two adjacent double perovskite blocks (a double bilayer). Importantly, this is the first observation of db-PNRs in a layered perovskite system. In the a-type db-PNRs (Fig. 2b), the Ca/Sr atoms all displace in the same direction in the rocksalt sheet between two double perovskite blocks. In the b-type db-PNRs (Fig. 2c), we observe alternating left/right displacements of the Ca/Sr atoms. The a-type and b-type db-PNRs show displacement patterns similar to those of the CMO $A2_1am$ polar ground state structure viewed from the [010] and [100] zone axis, respectively (Fig. 1b). To quantify the structural distortions, we use an atom position refinement algorithm to accurately assess the atom positions and measure the center Ca/Sr atomic displacements in the double perovskite blocks[20,21]. The displacement is measured by comparing the positions of the center Ca/Sr atoms and the average positions of the top and bottom Ca/Sr atoms in the double perovskite blocks (see Supplementary Figs. 2 and 3). The displacement measurement is presented in Fig. 2b, c as colored vector maps superimposed on the ADF-STEM images of the db-PNRs. Figure 2b shows the a-type db-PNRs with a relative displacement of the Ca/Sr atoms of close to 40 pm, and Fig. 2c shows the b-type db-PNRs with an alternating left/right displacement of the Ca/Sr atoms of approximately 20 pm. Additionally, as illustrated in the vector map, the center Ca/Sr atoms exhibit much larger displacements than the matrix of the crystal (close to 0 pm as indicated by purple color, also see Supplementary Fig. 4). The strain mapping from our geometric phase analysis (GPA) overlaid on the ADF-STEM image (Fig. 2d, e) shows a large strain of around 5–10% along the [001] direction. This local strain is likely the cause of the diffraction contrast in the lower magnification ADF-STEM image in Fig. 2a. The schematics of the a-type and b-type db-PNRs are superimposed on the magnified ADF-STEM image and are shown in Fig. 2f, g.

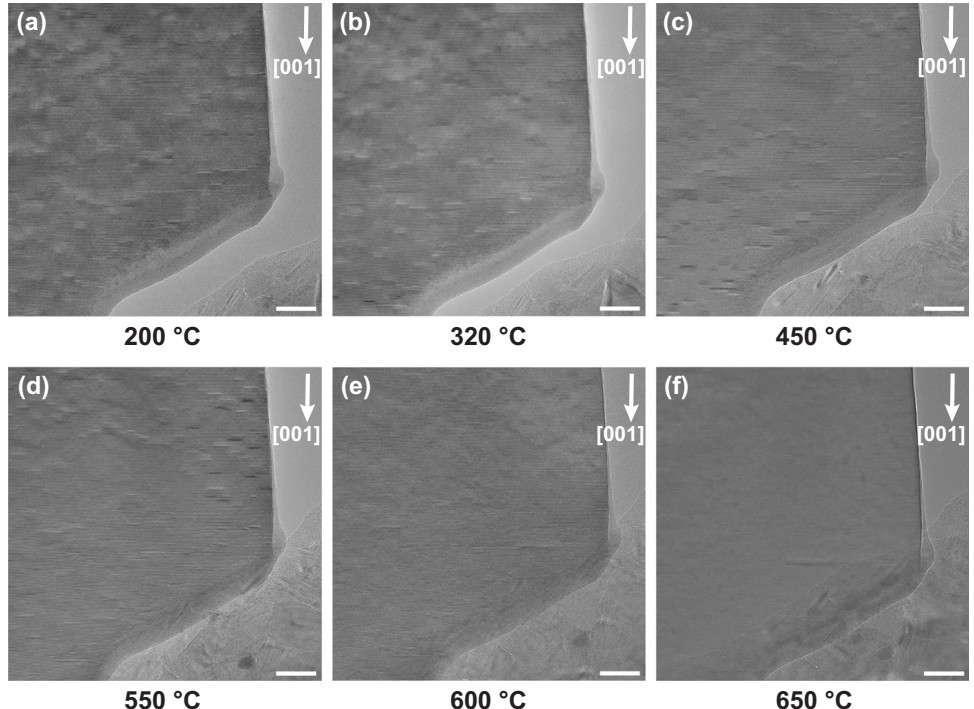

**Fig. 3 | TEM micrographs of the morphology of the CSMO sample taken from [100] zone axis at different temperatures during the in-situ heating experiment. a–f** The TEM image acquired at the temperature of 200 °C, 320 °C, 450 °C, 550 °C, 600 °C, and 650 °C, respectively. The [001] direction of the sample is labelled using the white arrow. The polar nanoregions can be seen as the randomly distributed, linear features that are perpendicular to the [001] direction. The white scale bar is 50 nm.

To investigate the possibility of oxygen vacancies[22,23] as well as to quantify the magnitude of the octahedral tilts, we make use of ABF-STEM. The ABF-STEM mode has been widely used for imaging light elements in a wide range of materials and is especially useful in visualizing oxygen octahedral distortions in oxide materials[24–26]. In this work, our ABF-STEM images uncover the oxygen atom columns in CSMO and show an intact crystal structure at and away from the db-PNRs regions, as shown in Fig. 2h. There is no clear indication of oxygen vacancy at atomic fractions above 40%, as it would significantly modulate the intensity of the oxygen atom columns locally (Supplementary Fig. 5)[27]. Additionally, the ABF-STEM images clearly illustrate a negligible tilt of the oxygen octahedra in the matrix as expected in the nonpolar *A*caa phase, whereas significant octahedral tilting is observed in the vicinity of the db-PNRs. We measure the oxygen octahedra tilting angle to be 158 degrees on average with a standard deviation of 4.7 degrees locally in the vicinity of the db-PNRs, as opposed to 178 degrees on average with a standard deviation of 1.6 degrees in the matrix.

To further understand the dynamics and stabilization of polar nanoregions in CSMO, we perform in-situ heating experiments inside the TEM column. Figure 3 shows the TEM micrographs of the morphology in the CSMO sample at different temperatures during heating. Initial heating of the sample starts from room temperature and goes to 200 °C, as shown in Fig. 3a. As discussed previously, the densely populated, nanometer-sized features are the db-PNRs. During heating from 200 °C to 450 °C, the location and the density of the db-PNRs are relatively stable and do not show much change as shown in Fig. 3a–c. Within this temperature range, the sample undergoes strain relaxation, and the diffraction contrast induced by strain is reduced. As the temperature is further raised to 550 °C, the db-PNRs start to disappear in the nonpolar phase (Fig. 3d). By 600–650 °C, most of the db-PNRs undergo a phase transition into the nonpolar *A*caa phase (Fig. 3e, f). However, there are still a small number of db-PNRs remaining at 650 °C. Upon cooling, the db-PNRs start to reappear below 568 °C, and the densely populated db-PNRs are mostly recovered when cooling to room temperature (Supplementary Fig. 6), which shows a reversible phase transition. The oxygen vacancies may emerge during heating in the vacuum, but the reversibility of the formation of db-PNRs during the in-situ heating experiment suggests that the contrast change is less likely to be linked to the oxygen vacancies[28]. This observation demonstrates a reversible and gradual transition between the polar and nonpolar phases in CSMO, but the mechanism that stabilizes db-PNRs at high temperatures is still unknown.

To understand the underlying physics of the db-PNR stabilization mechanism, we use monochromated EELS to determine the local chemical environment of the Mn ions in the db-PNRs. The L ionization edges of the transition metal elements (Mn in this case) reflect the electronic transition from 2p to 3d levels, with the $L_3$ and $L_2$ edges showing the transition from $2p^{3/2}$ to $3d^{3/2}$ $3d^{5/2}$ levels and from $2p^{1/2}$ to $3d^{3/2}$ levels respectively[29–32]. By performing monochromated EELS, the energy loss near edge structure (ELNES) of Mn $L_{2,3}$ is resolved for fingerprinting the oxidation states across the db-PNRs in the crystal. The ELNES for Mn with different oxidations states show distinct shapes and varied onset energy[33,34]. On the other hand, the ELNES for Mn with the same oxidation states and different bonding environment has similar overall shapes and onset energy[33]. Since the overall shape of the ELNES for the same Mn oxidation state in different compounds is similar, it has been successfully demonstrated that mixed Mn oxidation states can be determined with very high resolution by fitting the experimental Mn ELNES spectra with the existing reference ELNES spectra[34,35]. Figure 4a is a plot of the line-scan EELS data. In Fig. 4b, the corresponding spatial locations of the electron probe are plotted on the simultaneously acquired ADF-STEM image. Figure 4c shows the averaged EELS Mn L edge spectra from the db-PNRs (red) and nonpolar matrix (blue). We perform linear combination fitting on our line-scan EELS data set; the fitting of the averaged spectra from the polar and nonpolar regions is shown in Fig. 4c. We use the ELNES spectra from MnO ($Mn^{2+}$) and $SrMnO_3$ ($Mn^{4+}$) collected by Garvie et al.[33] as the

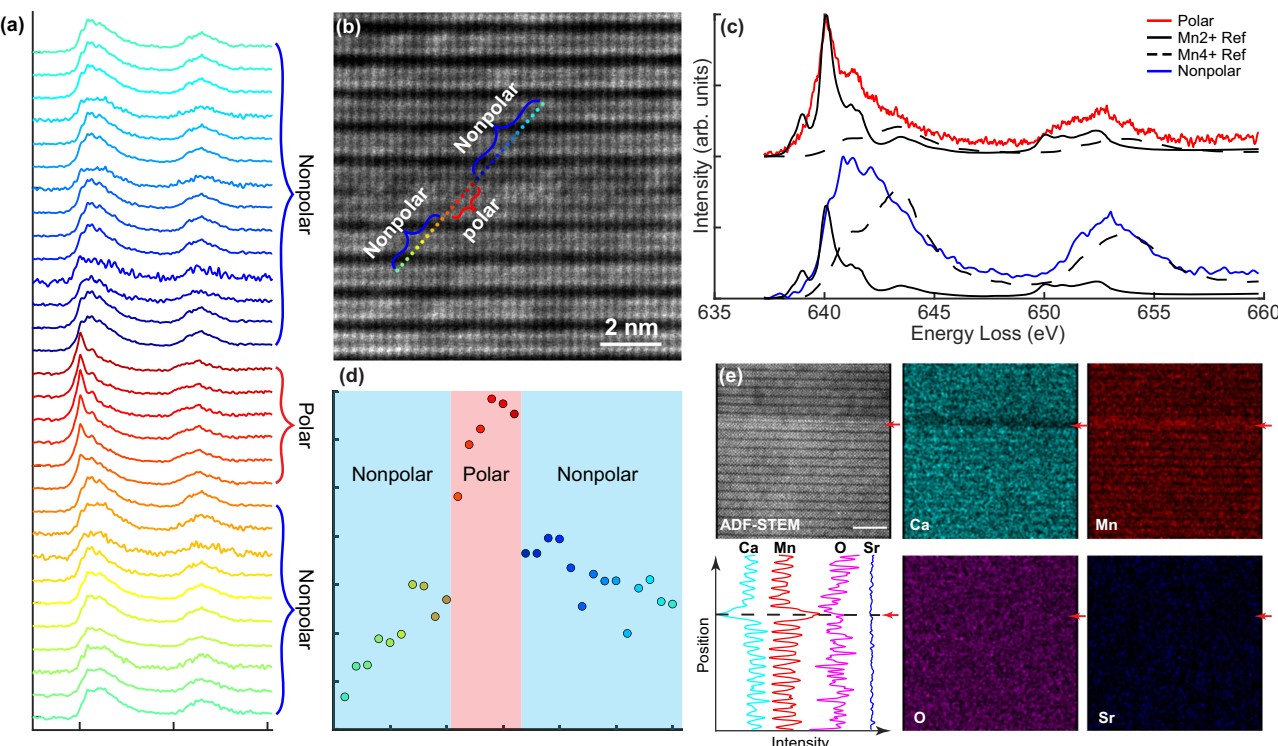

**Fig. 4 | Chemical environment study near db-PNRs. a** A plot of a line-scan EELS data across the db-PNR. The red and blue brackets on the right indicate the spectra from polar and non-polar regions, respectively. **b** The electron probe spatial locations superimposed on the ADF-STEM image. The red and blue brackets indicate the spectra from polar and non-polar regions, respectively. The color of the probe locations corresponds to the EELS spectra in **a**. **c** The averaged EELS Mn L edge spectra from the polar nanoregions (red) and non-polar matrix(blue), as indicated in **a**. The reference $Mn^{2+}$ and $Mn^{4+}$ spectra are also plotted as black solid and dashed lines. The height of the reference spectra is plotted based on the ELNES fitting results on both averaged spectra. **d** The plot of $Mn^{2+}$ percentage versus the probe positions from the ELNES fitting results for the entire line-scan EELS data. The red and blue background colors indicate the polar and nonpolar regions, respectively. A spike in $Mn^{2+}$ percentage is shown at the center of the polar region. **e** The EDX mapping near the db-PNR region. The approximate location of the db-PNR is pointed by the red arrows. The line profile of the EDX mapping intensity highlights the locally decreased and increased concentration of the Ca and Mn at db-PNR, respectively.

reference spectra. The fitting results indicate that the ELNES of the Mn L-edge at the db-PNRs shows more similarity to the $Mn^{2+}$ reference spectra. The fitting results of the entire line-scan EELS data are plotted (Fig. 4d) and indicate that the $Mn^{2+}$ percentage is approximately 90% at the center of the db-PNRs.

To confirm this result, we also determine the integrated intensity ratio of the $L_3/L_2$ edges of the line-scan dataset acquired from another region near the db-PNRs and obtain a ratio of approximately 2.6 and 2 in polar and nonpolar regions, respectively (Supplementary Fig. 7). The $L_3/L_2$ edge intensity ratio confirms that the oxidation state of Mn ions at the db-PNRs changes from the $Mn^{4+}$ state to either $Mn^{3+}$ or a mixture of $Mn^{2+}/Mn^{4+}$ states[29]. Additionally, we observe a much stronger Mn L edge intensity from the center rocksalt layer of the db-PNRs compared to the perovskite layer (Supplementary Fig. 8). Energy-dispersive X-ray spectroscopy (EDX) mapping of the same region confirms a higher Mn ion and a lower Ca ion content in these rocksalt sheets (Fig. 4e). Our observations indicate that Mn antisite defects form at the rocksalt layer Ca site of the Ruddlesden-Popper structure. The multi-valent Mn ions change oxidation states when forming antisite defects on the Ca sites.

To validate our hypothesis that Mn antisite defects stabilize the db-PNRs in CSMO, we perform density functional theory (DFT) + U calculations on $Ca_{3-x}Mn_{2+x}O_7$ for a range of x. We do not include the Sr dopants in our calculations to keep our computational cell to a reasonable size. The main effect of this choice is that $A2_1am$ is the ground state structure for all Mn dopant concentrations that we consider, however, the energy difference $\triangle E = E_{Acaa} - E_{A2_1am}$ can still inform us about how the substitution of Mn for some Ca cations

impacts the relative energy of the two phases. Figure 5a shows $\triangle E$ as a function of Mn-dopant concentration x for $Ca_{3-x}Mn_{2+x}O_7$, obtained from DFT + U structural relaxations. As the dopant concentration x increases, the energy difference $\triangle E$ grows, indicating that the $A2_1am$ structure is further stabilized with respect to $Acaa$. This result supports the interpretation of the experimental data that regions with Mn dopants stabilize the polar structure with $A2_1am$-symmetry.

Figure 5b–g show the Mn-O-Mn bond angles for several Mn-dopant concentrations, obtained from our DFT + U structural relaxations. The structures in Fig. 5(b–g) depict the lowest energy dopant configuration for each doping level. Other Mn-dopant configurations are shown in Supplementary Figs. 10 and 11. The $n = 2$ Ruddlesden-Popper $A_3B_2O_7$ structure contains two symmetry-distinct A-sites: a larger A-site situated in the center of the perovskite bilayer, and a smaller A-site located in the rock salt layer. Our DFT calculations show that it is always energetically favorable to place the smaller $Mn^{2+}$ dopants in the rocksalt A-sites, which agrees with prior experimental observations. Interestingly, when more than one dopant is present in the supercell (Fig. 5c–f), we find that it is energetically favorable for the $Mn^{2+}$ dopants to cluster in a single rocksalt layer. We also find that as the Mn-dopant concentration increases, the bond angles bend further from 180°. This effect can be rationalized by considering the tolerance factor: a decrease in the A-site cation ionic radius decreases the tolerance factor, which leads to increased octahedral rotation angles[36,37]. The Shannon radii of $Ca^{2+}$ and $Mn^{2+}$ (in 8-fold coordination) are 1.12 and 0.96 respectively. Thus, as the $Mn^{2+}$ dopant concentration

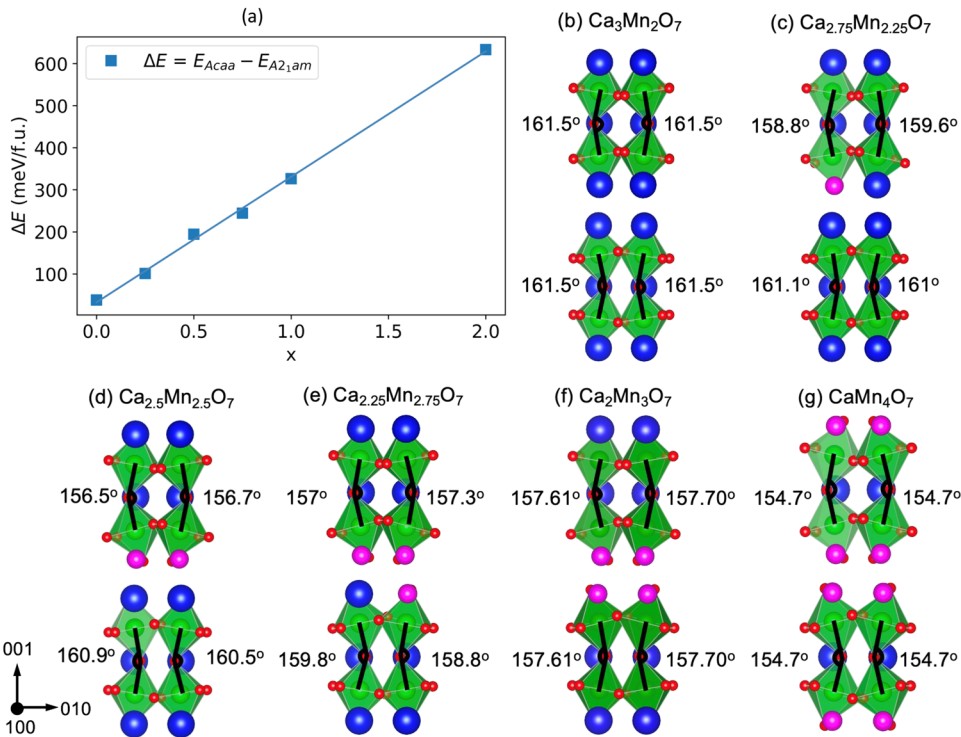

**Fig. 5 | DFT calculations. a** Energy difference between the nonpolar $A$caa and polar $A2_1$am phases of $Ca_{3-x}Mn_{2+x}O_7$ as a function of Mn dopant concentration x calculated with DFT + U. For Mn concentrations where there is more than one possible configuration of Mn dopants, we report the energy of the lowest energy configuration (the energies of other configurations are shown in the SI). **b**–**g** Mn-O-Mn bond angles for a range of $Ca_{3-x}Mn_{2+x}O_7$ compositions. For each composition, the lowest energy dopant configuration is shown. Blue, green, magenta, and red balls represent calcium, $Mn^{4+}$, $Mn^{2+}$, and oxygen atoms, respectively.

## Table 1 | The magnetic moment of Mn-dopant atoms in $Ca_{3-x}Mn_{2+x}O_7$ for structures with symmetry $A2_1$am and $A$caa calculated with DFT + U

| Material | x | Magnetic moment, $A2_1$am ($\mu B$) | Magnetic moment, $A$caa ($\mu B$) |
|---|---|---|---|
| $Ca_{2.75}Mn_{2.25}O_7$ | 0.25 | 4.40 | 4.45 |
| $Ca_{2.5}Mn_{2.5}O_7$ | 0.5 | 4.37 | 4.43 |
| $Ca_{2.25}Mn_{2.75}O_7$ | 0.75 | 4.37 | 4.42 |
| $Ca_2Mn_3O_7$ | 1.0 | 4.35 | 4.41 |
| $CaMn_4O_7$ | 2.0 | 4.34 | 4.41 |

For compositions with more than one dopant atom in the supercell, the reported magnetic moment is obtained by averaging the moments of all the dopant atoms in that cell. The magnetic moment of the B-site Mn atoms (that is, those within the oxygen octahedra) is ~2.6–2.7 $\mu B$ for all structures, which is consistent with a $Mn^{4+}$ charge state.

increases, the tolerance factor decreases, and the octahedral tilting increases (more bending of the Mn-O-Mn bond). We further notice that the bond angles closest to the $Mn^{2+}$ dopants change the most compared to their values in the undoped compound in Fig. 5b, whereas the bond angles further away from the dopant change a smaller amount.

We also use our DFT + U calculations to probe the charge state of the Mn dopants substituted on the Ca sites. In $Ca_3Mn_2O_7$, the Mn cations have a formal charge of 4+, whereas we expect a formal charge of 2+ if Mn substitutes onto a Ca site. $Mn^{2+}$ can adopt high- or low-spin states, which would correspond to magnetic moments of 5 $\mu B$ and 1 $\mu B$, respectively. Table 1 reports the magnetic moments of the Mn-dopants obtained from our DFT + U calculations. We find that for all dopant concentrations, the moment of the Mn dopants is 4.3–4.4 $\mu B$, which is consistent with the $Mn^{2+}$ charge state in the high-spin configuration. This agrees with the experimental observation of the existence of the $Mn^{2+}$ oxidation state in the polar region. Overall, the DFT

calculations exhibit excellent agreement with our experimental observations from atomic resolution STEM combined with EELS.

In conclusion, our study uncovers that the coexistence of polar $A2_1$am and nonpolar $A$caa phases in CSMO leads to the formation of db-PNRs. PNRs are widely viewed as the embryo of the ferroelectric phase and are critical to the relaxor properties of ferroelectrics in ultrasonic applications due to their superior piezoelectric properties[38], and the discovery of db-PNRs in a hybrid improper ferroelectric system is rare. The interpretation of the origin of the polar nanoregions in the relaxor ferroelectric materials has been challenging because of the heterogeneity over different time and length scales[39–41]. By employing atomic-resolution STEM imaging in combination with EELS and DFT calculations, this study demonstrates the polar $A2_1$am phase stabilization mechanism of db-PNRs in CSMO to be the formation of the Mn antisites on the Ca sites that increases the octahedral tilting amplitudes. This study utilizes an in-situ heating experiment to further explore the polar/nonpolar phase transition as a function of temperature and observed the presence of the db-PNRs at temperatures as high as 650 °C. Both previous studies[19] and our in-situ heating experiment show that the polar $A2_1$am and nonpolar $A$caa phases in CSMO exhibit a competition over a large temperature range, but the stabilization mechanism has not been clear until now. The stabilization mechanism of db-PNRs in CSMO due to Mn antisites is similar to Ti antisites producing PNRs and a switchable polarization in $SrTiO_3$[42,43], as well as the recent report of Y antisites leading to room temperature ferroelectricity in yttrium orthoferrite $YFeO_3$[44]. Our work shows that antisite defects play an important role in stabilizing polar nanoregions in the family of layered perovskite crystals and beyond. This study provides a path toward engineering polar nanoregions and designing novel lead-free relaxor ferroelectrics in hybrid improper ferroelectric materials by tuning the stoichiometry during growth for sophisticated defect engineering.

## Methods

### Computational details

We perform density functional theory (DFT) calculations using the Vienna Ab initio Simulation Package (VASP)[45,46]. We employ the PBEsol exchange-correlation functional and a plane-wave basis with an energy cutoff of 600 eV[47]. All calculations are performed in a 48-atom computational cell with a $6 \times 6 \times 2$ Monkhorst-Pack k-point mesh to sample the Brillouin zone. For structural relaxations, a force convergence tolerance of 2 meV/A is used, and both lattice parameters and atomic positions are optimized. We make use of the Liechtenstein formulation of the DFT + U method[48] to treat the Mn on-site Coulomb interaction and set the Coulomb and exchange parameters to be $U = 4.5$ eV and $J = 1.0$ eV, respectively, in agreement with previous work on $Ca_3Mn_2O_7$[2]. We have checked that our results are robust against reasonable variations of the U parameter. In addition, we note that in principle describing the A-site $Mn^{2+}$ dopant atoms may require a different value of U compared to the $Mn^{4+}$ atoms. We have checked that our results are robust against varying the U on the $Mn^{2+}$ dopant atoms (while keeping $U = 4.5$ eV on the $Mn^{4+}$ atoms). We impose G-type antiferromagnetic order in the perovskite bilayers, which is the known ground state magnetic structure of $Ca_3Mn_2O_7$[2]. For compositions with Mn dopants substituted on some Ca sites, the Mn dopant spins are placed in a ferromagnetic order. We have checked that selecting an antiferromagnetic order for the Mn dopant spins does not produce a qualitative change to our results. The DFT + U-relaxed lattice parameters of all $Ca_{3-x}Mn_{2+x}O_7$ compositions discussed in the main text are reported in Supplementary Table 1. We note that due to the ordered nature of the Mn dopant atoms in the periodic computational supercell, for most dopant concentrations the symmetry of the crystal structure is lower than the $A2_1am$ or $Acaa$ symmetry of $Ca_3Mn_2O_7$. However, for simplicity we choose to refer to these Mn-doped structures by the symmetry of the corresponding $Ca_3Mn_2O_7$ structure ($A2_1am$ or $Acaa$) throughout the text. We use the ISOTROPY Software Suite[49] for group-theoretic analysis and VESTA[50] to visualize crystal structures.

### Sample preparation

The high quality polycrystalline CSMO crystals were first prepared with the solid state reaction method by mixing the powders of $CaCO_3$ (99.99%, Alfa Aesar), $SrCO_3$ (99.99%, Alfa Aesar), and $MnO_2$ (99.997%, Alfa Aesar) in the stoichiometric ratio, and then pelletized and heated at 1350–1650 °C for 200 h in oxygen flow. Single-crystalline CSMO were grown by using optical floating zone growth methods (also see refs. 7, 19). The TEM samples were prepared using the Thermo Scientific Helios NanoLab Dual-Beam Focused Ion Beam (FIB) system. Before mounting the CSMO crystal into the FIB/SEM column, a thin layer of carbon was deposited for better electron conductivity and preventing charging. The regular TEM samples were prepared by first lifting out the 1–2 μm lamella from the crystal surface and mounting the lamella onto the copper V-post TEM grid. The mounted lamella was then thinned down using the 30 kV Ga ion beam to around 400 nm and subsequently to around 100 nm (electron transparent) using the 5 kV Ga ion beam. Later, 2 kV and 1 kV ion beams were applied to gently reduce the amorphization and Ga ion implantation caused by Ga ion bombardment.

The in-situ TEM samples were prepared by an ex-situ lift-out method. First, we created a trench at the crystal surface with the lamella sitting vertically in the center. Second, the lamella was thinned down to around 100 nm using the 30 kV and 5 kV ion beam. Third, we used the 2 kV ion beam to mill the sample surface for final cleaning. Then, the arms of the lamella that attaches to the trench were cut off and this step makes the electron transparent sample left free-standing in the trench. Subsequently, the crystal was carefully taken out of the SEM column and transferred to the EXpressLO workstation. Lastly, we used a glass needle with a very fine tip to lift out the sample using

electrostatic force and placed the sample flat onto the windows in the SiC membrane of the Protochips heating E-chip.

### Transmission electron microscopy

The aberration-corrected scanning transmission electron microscopy (STEM) images and the electron energy loss spectroscopy (EELS) spectra were both collected using the Thermo Scientific Titan[3] S/TEM equipped with a spherical aberration corrector, a monochromator, and a Gatan imaging filter (GIF). The STEM imaging was operated at the acceleration voltage of 300 kV with the probe convergence angle of 30 mrad. The ADF/ABF STEM images are acquired using HAADF and ABF detectors, respectively. The collection angle for the ADF detector is 44–244 mrad while for the ABF detector is 9–51 mrad. During the STEM-EELS data acquisition, the acceleration voltage is reduced to 80 kV for alleviating the beam damage. The EELS data was acquired in dual-EELS mode, in which each EELS data set consists of the low-loss spectra containing the zero-loss peak (ZLP) and the high-loss spectra with the core-loss edges. The collection angle for EELS data acquisition is 9.2 mrad. After activating the monochromator, the aberration correction is performed to enhance spatial resolution. The energy resolution is approximately 0.15 eV by measuring the FWHM of the ZLP. The exposure time for each probe position is 4 s.

The in-situ heating experiment was performed inside the Thermo Scientific Talos F200X TEM. The Protochips heating E-chip was first mounted onto the Protochips Fusion Select holder with the double tilt capability. After the holder was inserted into the TEM column, we first used the double tilt function to find the correct zone axis for the observation, and then start the heating process at the rate of 10 °C per minute until the temperature reached set goals. Several TEM images were captured during the heating process. After the sample reached a certain temperature, the sample position and tilt angles were adjusted to ensure the image to be acquired from relatively the same region, and then the temperature was held for around 10–15 min for the potential dynamics to take place. Multiple TEM images were taken at different magnifications every one to two minutes. The TEM images in Figs. 1 and 3 are acquired without inserting the objective aperture and at a slight defocus to enhance the contrast.

### STEM and EELS analysis

Each STEM image analyzed and presented in this work is drift corrected with a non-linear drift correction algorithm, with the input being a pair of sequentially acquired STEM images with perpendicular scanning directions[51]. The atom position analysis and displacement measurement were performed using the customized Matlab code[21,52]. The atom positions in ADF/ABF STEM images were determined by fitting a 2D-gaussian over the atomic columns and we defined the refined atomic positions as the center of the Gaussian peak. The displacement is measured by comparing the positions of the center Ca/Sr atoms and the average positions of the top and bottom Ca/Sr atoms in the double perovskite blocks (see Figs. S2 and S3). The GPA strain measurement was performed using Strain ++ software[53,54]. Prior to the analysis, a Hann window was applied on the STEM image input to reduce the fast Fourier transform (FFT) artifact. Then we applied FFT to the input image and obtained the power spectrum. Afterward, the Bragg-filtered images were generated by first applying two gaussian masks over the first Brillouin zone of two reciprocal lattice vectors and then perform an inverse Fourier transform. Later the phase images can be extracted from the Bragg-filtered images and the strain field can be determined from the phase images.

Before analyzing the EELS line-scan data, all energy loss axes in the high-loss spectra were first corrected according to the shift in the ZLP positions measured in the simultaneously collected low-loss spectra. The corrected high-loss EELS spectra were then denoised by employing the principal component analysis method and select the first five principal components to reconstruct the line-scan EELS data[55]. For the

EELS white-line ratio calculation, we first subtract the continuum contribution using a Hartree–Slater cross-section and determined $L_3/L_2$ intensity by fitting Gaussian peaks and integrating the intensity within the fitted Gaussian peaks (Supplementary Fig. 12)[29]. The ELNES fitting was performed on monochromated EELS spectra and the oxidation state was obtained by fitting the Mn L edges as a linear combination of two reference spectra of different oxidation states. The spectra are adjusted to accommodate the different calibration of the drift tube in different TEM instrument.

## Data availability

Relevant data supporting the key findings of this study are available within the article and the Supplementary Information file. All raw data supporting the findings of this study within the article, as well as the Supplementary Information file, are available from the corresponding author upon request.

## Code availability

The code to generate the atom position finding and refinement in STEM images based on the EASY-STEM MATLAB app is available at https://github.com/miaoleixin1994/EASY-STEM. The Matlab code used for EELS data analysis is available upon request.

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

## Acknowledgements

N.A. and L.M. acknowledge the support by NSF through the Pennsylvania State University Materials Research Science and Engineering Center (MRSEC) DMR-2011839 (2020–2026). Crystal growth at Rutgers was supported by the center for Quantum Materials Synthesis (cQMS), funded by the Gordon and Betty Moore Foundation's EPiQS initiative through grant GBMF10104, and by Rutgers University. K.H. and E.A.N. acknowledge support from University of California, Merced. L.M. and P.M. acknowledge support from NSF DMR-1420620. We acknowledge the use of computational resources provided by the Extreme Science and Engineering Discovery Environment (XSEDE), which is supported by National Science Foundation Grant No. ACI-1548562, as well as the Multi-Environment Computer for Exploration and Discovery (MERCED) cluster at University of California, Merced, which is supported by National Science Foundation Grant No. ACI-1429783. D.M. was supported by ORNL's Laboratory Directed Research and Development (LDRD) Program, which is managed by UT-Battelle, LLC, for the U.S. Department of Energy (DOE) under the contract no. DE-AC05-00OR22725. We appreciate the help from Dr. Rongwei Hu with crystals growth and Dr. Fei-Ting Huang for insightful discussions. We appreciate the insight and suggestion from Prof. Venkatraman Gopalan. We appreciate the support and resources from Materials Characterization Lab (MCL) at Penn State.

## Author contributions

L.M. designed the study in consultation with N.A. L.M. carried out the in-situ TEM, STEM/EELS experiments and data analysis. P.M., D.M. and K.W. contributed to the TEM data collection. K.H. and E.A.N. carried out the DFT simulations and S.W.C. carried out the crystal synthesis. The manuscript was drafted by L.M. and edited by N.A. The manuscript was written through contributions of all authors. All authors have given approval to the final version of the manuscript.

## Competing interests

The authors declare no competing interests.
