## [Peer Review File · Nature Communications]

REVIEWER COMMENTS

Reviewer #1 (Remarks to the Author)

The manuscript “Double-Bilayer Polar Nanoregions and Mn Antisites in $(\text{Ca}, \text{Sr})_3\text{Mn}_2\text{O}_7$ ” by Leixin Miao et al. uses a combination of several experimental and first-principles techniques to study polar nano-regions in the nonpolar phase of $(\text{Ca}, \text{Sr})_3\text{Mn}_2\text{O}_7$.

I find the manuscript interesting, well-written, with easy-to-follow text and progression. The techniques employed are appropriate and reasonable conclusions are drawn from experimental data and supported by first-principles theory. Progress towards both room-temperature multiferroics, new ferroelectric, and further understanding of negative thermal expansion materials are important endeavors, where new strategies for enhancement control are always of interest.

I believe this work will be appropriate for publication in Nature Communications after several modest changes (particularly to the presentation of figures) suggested to help the audience follow the discussion.

Suggestions

The use of acronyms is unnecessarily heavy. I suggest backing off of their use so that the reader can focus on following the scientific development presented.

Figure 1

Incorporation of Glazer notation and showing important tilt patterns would be helpful since they are discussed in the main text and caption.

Scale bar should be made clear in (d)

Figure 2

I suggest adding a schematic of the a-type and b-type db-PNRs. In the current manuscript, the reader is expected to translate technical language into a picture.

Labeling false-color legends would be helpful.

Figure 4

Red and blue brackets should be labeled as “polar” and “non-polar” regions in the figure

“The results of the ELNES fitting results.” Perhaps omit the second results.

Color bar should be labeled.

Polar and nonpolar regions should be marked in (c).

Reviewer #2 (Remarks to the Author)

Atomic scale observation of PNR in non-polar matrix, first observation of db-PNR in RP perovskite. This paper observes the PNRs with atomic resolution STEM and inspects their local chemistry with EELS and EDS to find a prevalence of Mn²⁺ at the PNR in combination with an excess of Mn and depletion of Ca. The paper interprets the origin of the PNRs as a result of stabilization of Mn antisite defects on the Ca site, driving octahedral tilting and polarity, through DFT studies. I believe the work will be of significance to the field of functional materials, and of most interest to those in improper ferroelectrics, relaxor ferroelectrics, and multiferroics. My overall recommendation is that the work should be published, taking into consideration some comments on the STEM data analysis and presentation which are listed below.

1. Please include details on the TEM image acquisition for Fig 1d and Figure 3. In particular, what objective aperture was used to obtain this contrast? Which direction is [001] (if known) for Fig 1d?
2. Figure 1e. The text is too small labeling the spots, and if you zoom up to read it, there is a typo in the indexing.
3. Figure 2b-c and lines 118-120. From what you describe it seems Figure 2b should be 2c and 2c should be 2b. Looking at the atoms reading from left to right, the atoms in Figure 2b are all displaced in the same direction, and that those in 2c are alternating left/right. Right now the text in lines 118-120 says the reverse. Please correct or clarify.
4. The arrows in Figure 2b-c are not clearly defined (in this paper nor the reference). From reading reference 20, I would anticipate that you are plotting the net polarization displacement, but how that is defined/measured is not so clearly stated. In the methods, you state, “The atomic displacement was calculated by measuring the distance from the displaced atoms to the unit cell center positions.” How do you measure the unit cell center position – what is your reference? Also is the displaced atoms just the center Ca atom (as inferred from your figure caption)? Why are you plotting the displacement of the center Ca ion instead of a column average, or the ones near the fault, which are doing the most interesting behavior?
5. Figure 2e. I cannot directly observe the oxygen tilting in Figure 2e due to the annotations of the tilting. I think the best practice is to include atomic resolution images from Figure 2 without atomic annotations in the SI so that we can look at the raw data as well.
6. On line 140-141, the authors make the claim that, “There is no clear indication of oxygen vacancy ordering as it would modulate the intensity of the oxygen atom columns locally.” ABF-STEM is notorious for having difficult to interpret intensities, and I doubt it is sensitive to oxygen vacancies at low concentrations. How many oxygen vacancies would you have to have before ABF showed you

something? I have only seen work showing that this is possible for vacancy concentrations between 25-40% of atomic columns. You could perhaps say, "There is no clear indication of oxygen vacancy ordering at atomic fractions above X% as it would modulate the intensity of the oxygen atom columns locally.[REF]" but as it stands this statement is doubtful.

7. Lines 145-146. Please state the mean and standard deviation of the octahedral tilt measurement in the non-polar matrix. Right now it is listed as "approximately 180 degrees."

8. About the in-situ TEM heating experiment. It is well known that oxides will lose oxygen at high temperatures in the vacuum of the microscope. I would expect oxygen vacancies and oxygen deficiency in the structure after this heating experiment. Some comment on the possibility of losing oxygen is needed, otherwise I may think that the contrast change is due to some oxygen effect. Some other questions, when you cool the sample down, do the polar regions re-appear, or has the sample undergone a permanent phase transition? Are there changes in the O-K ELNES, or other spectroscopy near the defects (do the antisites heal)?

9. Right now I'm not sure what is learned from the in situ heating measurement other than it is a gradual transition, which seems already known. Maybe the community is interested in seeing this regardless but it does seem less informative and less relevant than other aspects of the paper, regarding the PNR and Mn antisites.

10. Figure 4. Please plot the image and the EELS separately, I found it very difficult to observe the EELS data, or have the EELS data separately in the SI.

11. I find it surprising that the Mn ELNES shape is the same regardless of crystal symmetry – in fact, reference 32 states, "In contrast, the Mn L3-edges from Mn³⁺ and Mn⁴⁺ containing minerals exhibited ELNES that are interpreted in terms of a crystal-field splitting of the 3d electrons, governed by the symmetry of the surrounding ligands." Which is more in line with my thinking. However, references 31 and 33 do seem to agree with your statement – although I suspected that onset energy or using a perovskite reference might be more reliable than shape fitting. It is good that you analyzed using a second method with the L3/L2 ratios, although edge intensity ratios tend to have multiple scattering issues (ideally thickness is not changing much). Because you have two methods and the precedence of references 31 and 33, I think this is okay, but I'm not sure how much help Ref 32 is doing you as it is for a different class of materials and the sentence above seems contradictory.

12. The cation concentrations as measured by EDS (and EELS) are important to the argument in the paper. Including a line profile of these in Figure 4 would be helpful to the reader. You have an interesting story about Mn antisites and PNRs, but it seems like some measurements that were more tangential (in situ heating, strain mapping) got priority in the manuscript over some of the more bread-and-butter tools (elemental mapping) which provide central elements to the story.

Reviewer #3 (Remarks to the Author)

In this manuscript the authors present an experimental (TEM and EELS) and theoretical (DFT calculations) on Ca_{2.9}Sr_{0.1}Mn₂O₇ layered perovskite related to the Ca₃Mn₂O₇ hybrid-improper ferroelectric (HIF) crystal. From the experimental side, it is observed that double bilayer polar nano regions (db-PNRs) are actually at play through the coexistence of polar (A₂_1am) and non-polar (Acca) phases and that their stabilization is directly associated to the presence of Mn²⁺ antisite

defects in the rock salt layer instead of the Mn⁴⁺ coming from the B-site position inside the octahedra. This observation is also supported by DFT calculations.

The paper is well written and the different conclusions are supported by several experimental measurements and data as well as DFT calculations.

Additionally, the fact that db-PNRs are key for the ferroelectric or relaxor properties in layered perovskites, I think, makes the overall research presented in this manuscript of interest as a publication in Nature Communication.

As I'm a theorist, I'll give mostly my comments regarding the DFT calculations.

Here are my comments that, I think, the authors should address before publishing the article

1) In page 11, line 221, the authors mention that the high- or low-spin state of Mn²⁺ correspond to a magnetic moment of 5 μ_B and 0.5 μ_B respectively. While I agree for 5 μ_B for the high spin, I would say that the low spin state should have a magnetic moment of 1 μ_B and not 0.5, right

2) In the computational method, the authors mention that they have used DFT+U method with one set of UJ parameter. How this specific value of UJ parameter was selected Another question is that they have Mn⁴⁺ and Mn²⁺ cations, which would require a different UJ parameters. Again how the authors decided about having the same UJ parameters for two different oxidization state Did they check that the results are not affected by changing the UJ parameters (I would say this is OK as the energy between structures are compared and these energy differences are larger than the magnetic ones, but this should be quickly proved that it is indeed the case)

3) The Mn²⁺ in the rock-salt layers are placed in a ferromagnetic order. Again, is it the ground state of the system Is antiferromagnetism or ferromagnetism does not change the qualitative picture reported in the manuscript (again, as previous comment, I would say not in view of the energies at play that are larger than the magnetic ones but it should be shown to be indeed the case)

Response to Reviewer 1

Reviewer #1 (Remarks to the Author):

The manuscript “Double-Bilayer Polar Nanoregions and Mn Antisites in (Ca, Sr)₃Mn₂O₇” by Leixin Miao et al. uses a combination of several experimental and first-principles techniques to study polar nano-regions in the nonpolar phase of (Ca,Sr)₃Mn₂O₇.

I find the manuscript interesting, well-written, with easy-to-follow text and progression. The techniques employed are appropriate and reasonable conclusions are drawn from experimental data and supported by first-principles theory. Progress towards both room-temperature multiferroics, new ferroelectric, and further understanding of negative thermal expansion materials are important endeavors, where new strategies for enhancement/control are always of interest.

I believe this work will be appropriate for publication in Nature Communications after several modest changes (particularly to the presentation of figures) suggested to help the audience follow the discussion.

The use of acronyms is unnecessarily heavy. I suggest backing off of their use so that the reader can focus on following the scientific development presented.

Response: We appreciate the reviewer for the positive comments. We appreciate the suggestion regarding the usage of the acronyms. We have reduced the use of acronyms in the manuscript to increase the clarity. In particular, we remove the use of NTE (negative thermal expansion), HREM (high-resolution electron microscopy), HIF (hybrid improper ferroelectric).

Figure 1:

Incorporation of Glazer notation and showing important tilt patterns would be helpful since they are discussed in the main text and caption.

Scale bar should be made clear in (d)

Response: We thank the reviewer for the suggestion. We have added the schematics of the oxygen octahedral tilting patterns with glazer notations to help the reader visualize the crystal structure in Figure 1a. The scale bar has been made more obvious in Figure 1d.

Figure 2:

I suggest adding a schematic of the a-type and b-type db-PNRs. In the current manuscript, the reader is expected to translate technical language into a picture.

Labeling false-color legends would be helpful.

Response: We appreciate the suggestion from the reviewer. We added the schematics of the a-type and b-type db-PNRs superimposed on the experimental ADF-STEM images in Figure 2b and 2c to help the readers visualize the crystal structure. We also added labels on all the scalebars in Figure 2.

Figure 4:

Red and blue brackets should be labeled as “polar” and “non-polar” regions in the figure

“The results of the ELNES fitting results.” Perhaps omit the second "results".

Color bar should be labeled.

Polar and nonpolar regions should be marked in (c).

Response: We appreciate the suggestions from the reviewer to help us improve the figure quality. We have modified the overall representation of the Figure 4 as described below:

1) We have plotted the EELS spectra and ADF-STEM image separately in Figure 4a and 4b to help readers examine the spectra and the image more easily.

2) we have labeled the red and blue brackets as “polar” and “nonpolar” in Figure 4a and 4b to help readers identify the spectra from each region.

3) we drew the scatter plot for the ELNES fitting for the Mn L edge in Figure 4d with the colors matching the colors of the probe positions in Figure 4b. The polar and nonpolar regions are marked by the red and blue background colors, respectively.

4) as suggested by the other reviewer, we added our EDS elemental mapping in Figure 4e to highlight the composition differences at the db-PNRs compared to the nonpolar matrix.

We appreciate the reviewer for pointing out the typo in the caption and it has been fixed.

Response to Reviewer 2

Reviewer #2 (Remarks to the Author):

Atomic scale observation of PNR in non-polar matrix, first observation of db-PNR in RP perovskite. This paper observes the PNRs with atomic resolution STEM and inspects their local chemistry with EELS and EDS to find a prevalence of Mn²⁺ at the PNR in combination with an excess of Mn and depletion of Ca. The paper interprets the origin of the PNRs as a result of stabilization of Mn antisite defects on the Ca site, driving octahedral tilting and polarity, through DFT studies. I believe the work will be of significance to the field of functional materials, and of most interest to those in improper ferroelectrics, relaxor ferroelectrics, and multiferroics. My overall recommendation is that the work should be published, taking into consideration some comments on the STEM data analysis and presentation which are listed below.

Response: We appreciate the reviewer for the positive comments, and we are grateful for the constructive suggestions.

1. Please include details on the TEM image acquisition for Fig 1d and Figure 3. In particular, what objective aperture was used to obtain this contrast? Which direction is [001] (if known) for Fig 1d?

Response: We appreciate the suggestions. The TEM images in Figure 1e (originally 1d) and Figure 3 are CTEM images acquired with a slight defocus to enhance the contrast without the objective aperture. For Figures 1 and 3, the following description is added in the methods section, line 353-354:

“The TEM images in Figure 1 and 3 are acquired without inserting the objective aperture and at a slight defocus to enhance the contrast.”

Additionally, the [001] direction is added and labeled with the white arrow in Figure 1e. The [001] direction is perpendicular to the linear features (db-PNRs).

2. Figure 1e. The text is too small labeling the spots, and if you zoom up to read it, there is a typo in the indexing.

Response: We appreciate the reviewer for pointing out. We enlarged the electron diffraction pattern in Figure 1f (originally Figure 1e) and increased the text size for the indexing.

Reviewer:

3. Figure 2b-c and lines 118-120. From what you describe it seems Figure 2b should be 2c and 2c should be 2b. Looking at the atoms reading from left to right, the atoms in Figure 2b are all displaced in the same direction, and that those in 2c are alternating left/right. Right now the text in lines 118-120 says the reverse. Please correct or clarify.

Response: We sincerely appreciate the reviewer for pointing out this typo in the manuscript and help us improve the clarity the point. Indeed as you pointed, in the a-type db-PNRs, the Ca/Sr atoms are all displaced in the same direction in the rocksalt sheet between two double perovskite blocks; and in the b-type db-PNRs, Ca/Sr atoms show alternating left/right displacements.

We have corrected our writing in lines 122 – 124:

“... In the a-type db-PNRs (Fig. 2b), the Ca/Sr atoms are all displaced in the same direction in the rocksalt sheet between two double perovskite blocks. In the b-type db-PNRs (Fig. 2c), we observe alternating left/right displacements of the Ca/Sr atoms.”

4. The arrows in Figure 2b-c are not clearly defined (in this paper nor the reference). From reading reference 20, I would anticipate that you are plotting the net polarization displacement, but how that is defined/measured is not so clearly stated. In the methods, you state, “The atomic displacement was calculated by measuring the distance from the displaced atoms to the unit cell center positions.” How do you measure the unit cell center position – what is your reference? Also is the displaced atoms just the center Ca atom (as inferred from your figure caption)? Why are you plotting the displacement of the center Ca ion instead of a column average, or the ones near the fault, which are doing the most interesting behavior?

Response: We appreciate the constructive comments from the reviewer on the atomic displacement measurements. We apologize that our previous description of the displacement measurement was not complete. For the measurement, we used the average positions of the Ca/Sr atoms at the top and bottom of the double-layer perovskite blocks as the reference. The plotted

vector maps showed the displacement of the center Ca/Sr atoms compared the reference position. We choose to use the average position of top and bottom Ca/Sr atoms as the reference point due to the polarization arises primarily from a two-against-one displacement of the Ca ions along the a-axis in each perovskite block (Figure 1b, right); the $a^-a^0c^0$ octahedral tilting distortion involves alternating left/right Ca/Sr atom displacements along the b-axis of the Ca ions in each layer (Figure 1b, left).

Consequently, the relative shift between the center Ca in the perovskite block and the average position of top and bottom Ca/Sr would reflect the two-against-one displacement of Ca ions along a-axis (highlighting the polarization displacement) and the alternating left/right displacement along the b-axis (highlighting the enhanced octahedral tilting). We added the description of the method of measurement in the manuscript line 128-130:

“... The displacement is measured by comparing the positions of the center Ca/Sr atoms and the average positions of the top and bottom Ca/Sr atoms in the double perovskite blocks (see Figure S2 and S3). ...”

We added Figure S2 and S3 in the supplementary information to help the reader understand our measurements method.

Figure S2.

Figure S3.

5. Figure 2e. I cannot directly observe the oxygen tilting in Figure 2e due to the annotations of the tilting. I think the best practice is to include atomic resolution images from Figure 2 without atomic annotations in the SI so that we can look at the raw data as well.

Response: We appreciate the suggestion from the reviewer for improving the presentation of our figures. We have modified the overall representation of the Figure 2 by

- 1) adding schematics of the a-type and b-type db-PNRs in Figure 2b and 2c to help the reader visualize the crystal structures.
- 2) we made modifications to Figure 2e based on the suggestion from the reviewer. We separated the magnified atomic resolution ABF-STEM image near the db-PNRs and the schematics highlighting the oxygen octahedral tilting to help readers examine the raw data. We also superimposed the crystal structure model over one double perovskite block in the image to assist the visualization of the structure locally.

6. On line 140-141, the authors make the claim that, “There is no clear indication of oxygen vacancy ordering as it would modulate the intensity of the oxygen atom columns locally.” ABF-STEM is notorious for having difficult to interpret intensities, and I doubt it is sensitive to oxygen vacancies at low concentrations. How many oxygen vacancies would you have to have before ABF showed you something? I have only seen work showing that this is possible for vacancy concentrations between 25-40% of atomic columns. You could perhaps say, “There is no clear indication of oxygen vacancy ordering at atomic fractions above X% as it would modulate the intensity of the oxygen atom columns locally.[REF]” but as it stands this statement is doubtful.

Response: We appreciate the reviewer’s comment. To further understand the impact of the potential oxygen vacancies on the contrast in the ABF-STEM image, we performed the STEM simulation with the Prismatic STEM simulation software. We have added the STEM simulation results in the supplementary information, Figure S5, which is also shown below.

To perform this study, we first constructed a $\text{Ca}_3\text{Mn}_2\text{O}_7$ model with A_{caa} space group symmetry and randomly removed oxygen atoms from a specific oxygen column in the model. We performed the simulation on the models with various oxygen vacancy percentages (Figure S5b). We have obtained the simulated ABF-STEM images by setting the collection angle to be 9-51 mrad, which is consistent with our experimental setup.

We then calculated the integrated intensity from a row of atomic columns (Figure S5c, inversed intensity) in the ABF-STEM image and plotted the line profile of the integrated intensity (Figure S5d, inversed intensity). We define that the intensity modulation is significant in the ABF-STEM image if the oxygen column intensity in the image drops by over 50 percent. The 50% drop in intensity is plotted as dashed lines in the line profile in Figure S5d for reference. We found that if the oxygen vacancy percentage surpasses 40%, the modulation would be significant as the oxygen column intensity would consistently drop below 50% threshold. As a result, as the reviewer suggested, we modified our claim in the manuscript in line 147-149:

“... There is no clear indication of oxygen vacancy at atomic fractions above 40% as it would significantly modulate the intensity of the oxygen atom columns locally (Figure S5).²⁷ ...”

We also cited the reference 27 for further support of our claim:

27. Ishida, Takafumi, Tadahiro Kawasaki, Takayoshi Tanji, and Takashi Ikuta. "Quantitative evaluation of annular bright-field phase images in STEM." *Microscopy* 64, no. 2 (2015): 121-128.

Ishida et al. showed that the oxygen column intensity in simulated ABF-STEM of SrTiO_3 drops by roughly 50% with the oxygen occupancy of roughly 60%, which agrees well with the ABF-STEM simulation we performed on the $\text{Ca}_3\text{Mn}_2\text{O}_7$ crystal model.

In addition, the EDS mapping in Figure 4e showed evidence for deficient Ca and excess Mn, whereas the oxygen mapping did not show significant variation.

7. Lines 145-146. Please state the mean and standard deviation of the octahedral tilt measurement in the non-polar matrix. Right now it is listed as “approximately 180 degrees.”

Response: We appreciate the reviewer for pointing out. We measured the tilting angle in the nonpolar matrix to be 178 degrees on average with the standard deviation of 1.6 degrees. We added the measurement result in the manuscript, lines 153-154:

“... as opposed to 178 degrees on average with a standard deviation of 1.6 degrees in the matrix.”

8. About the in-situ TEM heating experiment. It is well known that oxides will lose oxygen at high temperatures in the vacuum of the microscope. I would expect oxygen vacancies and oxygen deficiency in the structure after this heating experiment. Some comment on the possibility of losing oxygen is needed, otherwise I may think that the contrast change is due to some oxygen effect. Some other questions, when you cool the sample down, do the polar regions re-appear, or has the sample undergone a permanent phase transition? Are there changes in the O-K ELNES, or other spectroscopy near the defects (do the antisites heal)?

Response: We appreciate the reviewer’s comment. Admittedly, oxygen vacancy formation in vacuum under the in-situ heating conditions is possible. On the other hand, if the contrast change is primarily due to oxygen vacancy generated in vacuum, the transition of db-PNRs should not be reversible as the stoichiometry is undermined. To understand the reversibility of the transition, we recorded the morphology of the sample during the cooling process and included the data in supplementary Figure S6. We observed that the db-PNRs started to reappear after cooling below 568 °C, and the densely populated db-PNRs were mostly recovered after cooling down to room temperature. The cooling process during the in-situ experiment shows that the transition of the db-PNRs is reversible in vacuum, which shows the evidence of the limited contribution of the oxygen vacancies to the overall contrast change during the transition.

In the manuscript lines 165-170, we added the comment to the possibility of losing oxygen:

“... Upon cooling, the db-PNRs starts to reappear below 568 °C and the densely populated db-PNRs are mostly recovered when cooling to the room temperature (Figure S6), which shows a reversible phase transition. The oxygen vacancies may emerge during heating in vacuum, but the reversibility of the formation of db-PNRs during in-situ heating experiment suggests that the contrast change is less likely to be linked to the oxygen vacancies. ...”

9. Right now I’m not sure what is learned from the in-situ heating measurement other than it is a gradual transition, which seems already known. Maybe the community is interested in seeing this regardless but it does seem less informative and less relevant than other aspects of the paper, regarding the PNR and Mn antisites.

Response: We appreciate the reviewer's comment. The in-situ heating experiment demonstrated in this work is significant to our overall story because it not only confirms the previous reports of the polar/nonpolar phase coexistence and the gradual transition, but also uncovered that a portion of the db-PNRs could exist at high temperatures. The discovery led to the investigation into the chemical environment near the db-PNRs with EELS and EDX to further understand the stabilization mechanism behind the phase coexistence. In other words, the in-situ heating TEM experiment serves as a bridge between the structural characterization of the db-PNRs and the follow-up chemical environment characterization. Additionally, the heating experiment would draw some interests from the community for understanding the relationship between phase coexistence and negative thermal expansion properties in the hybrid improper ferroelectric system.

10. Figure 4. Please plot the image and the EELS separately, I found it very difficult to observe the EELS data, or have the EELS data separately in the SI.

Response: We are grateful for the reviewer's suggestions for improving clarity of the figures. We have modified the overall representation of the Figure 4:

1) we have plotted the EELS spectra and ADF-STEM image separately in Figure 4a and 4b to help readers examine the spectra more easily.

2) we have labeled the red and blue brackets as "polar" and "nonpolar" in Figure 4a and 4b to help readers identify the spectra from each region.

3) we have drawn the scatter plot for the ELNES fitting for the Mn L edge line scan data in Figure 4d with the colors matching the colors of the probe positions in Figure 4b. The polar and nonpolar regions are marked by the red and blue background colors, respectively.

4) we have added our EDS elemental mapping in Figure 4e to highlight the variation in the composition of the db-PNRs compared to the nonpolar matrix.

11. I find it surprising that the Mn ELNES shape is the same regardless of crystal symmetry – in fact, reference 32 states, "In contrast, the Mn L3-edges from Mn³⁺ and Mn⁴⁺ containing minerals exhibited ELNES that are interpreted in terms of a crystal-field splitting of the 3d electrons, governed by the symmetry of the surrounding ligands." Which is more in line with my thinking.

However, references 31 and 33 do seem to agree with your statement – although I suspected that onset energy or using a perovskite reference might be more reliable than shape fitting. It is good that you analyzed using a second method with the L3/L2 ratios, although edge intensity ratios tend to have multiple scattering issues (ideally thickness is not changing much). Because you have two methods and the precedence of references 31 and 33, I think this is okay, but I'm not sure how much help Ref 32 is doing you as it is for a different class of materials and the sentence above seems contradictory.

Response: We sincerely appreciate the reviewer for the constructive comments on the EELS analysis performed in the manuscript. We agree with the comments that the combination of ELNES fitting and L3/L2 ratios measurement supports our conclusion of Mn²⁺ formation. We would like to further discuss the usage of the references 32 and details of our EELS analysis.

1) For Mn⁴⁺, Garvie et al. (ref. 32) investigated the Mn⁴⁺ compounds with various arrangements of the Mn-O₆ octahedra (corner sharing, edge sharing or face sharing). The study found that the overall shapes and onset energy of the L edges for Mn⁴⁺ are very similar in different compounds, while the crystal field splitting mainly affect the fine structures within the L₃ peak. As a result, ref. 32 agrees with the ref 31 and 33. In conclusion, the since overall ELNES shape and onset energy are similar for Mn⁴⁺ compounds with octahedral coordination, the small difference in the fine shape of the L₃ edges does not affect the conclusion we have in the manuscript that the Mn L edge ELNES in db-PNRs is dominated by the Mn²⁺ component.

In the manuscript lines 179-182, we added the discussion on the ELNES similarities and differences for Mn in different compounds:

“The ELNES for Mn with different oxidations states show distinct shapes and varied onset energy. On the other hand, the ELNES for Mn with the same oxidation states and different bonding environment has similar overall shapes and onset energy.”^{33,34}

2) We appreciate the suggestion on the analysis method provided by the reviewer. In terms of the EELS analysis method selection, we have considered using onset energy shift measurement other than the ELNES fitting method and L3/L2 intensity ratio calculation we used in the manuscript. However, the onset energy shift measurement is prone to be dominated by the edges with lower onset energy.

In our EELS data, we observed that the peak position of the L_3 edges at the polar phase is shifted by around 2 eV towards the lower energy loss side, whereas the edge onset energy (inflection point of the rise of L_3 edge) is shifted by only roughly 0.3 eV. Since the L edge onset might have been heavily impacted by the mixed oxidation state near the db-PNRs, the edge onset measurement was not included in the manuscript.

12. The cation concentrations as measured by EDS (and EELS) are important to the argument in the paper. Including a line profile of these in Figure 4 would be helpful to the reader. You have an interesting story about Mn antisites and PNRs, but it seems like some measurements that were more tangential (in situ heating, strain mapping) got priority in the manuscript over some of the more bread-and-butter tools (elemental mapping) which provide central elements to the story.

Response: We appreciate the suggestions offered by the reviewer. We agree that the EDS mapping data serves an important role to establish our interpretation of the Mn antisites in the db-PNRs. As mentioned in our response to comment #10, we added the EDS elemental mapping and the intensity line profile to Figure 4e to help reader understand our interpretation.

Response to Referee 3

Referee:

1) In page 11, line 221, the authors mention that the high- or low-spin state of Mn^{2+} correspond to a magnetic moment of $5 \mu_B$ and $0.5 \mu_B$ respectively. While I agree for $5 \mu_B$ for the high spin, I would say that the low spin state should have a magnetic moment of $1 \mu_B$ and not 0.5 , right?

Response: The referee is correct that the magnetic moment of the low spin state should be $1 \mu_B$. We have corrected this in the text. We would like to thank the referee for finding this error.

Referee:

2) In the computational method, the authors mention that they have used DFT+U method with one set of U/J parameter. How this specific value of U/J parameter was selected? Another question is that they have Mn^{4+} and Mn^{2+} cations, which would require a different U/J parameters. Again how the authors decided about having the same U/J parameters for two different oxidization state? Did they check that the results are not affected by changing the U/J parameters (I would say this is OK as the energy between structures are compared and these energy differences are larger than the magnetic ones, but this should be quickly proved that it is indeed the case)?

Response: We selected U and J values that have been shown previously in the literature to accurately describe the structural and magnetic properties of $Ca_3Mn_2O_7$, see for example N. A. Benedek & C. J. Fennie, PRL 106, 107204 (2011). The reviewer is correct that Mn^{4+} and Mn^{2+} cations would in principle require different U/J parameters. However, previous work in the literature on MnO (which contains Mn^{2+} cations) report that a choice of $U-J = 4$ eV in the Dudarev implementation of DFT+U correctly captures the structural and magnetic properties, and that the structural parameters and magnetic moment are insensitive to variations of U by 1 eV (see for example A. Schron et al, PRB 82, 165109 (2010), A. Schron et al, PRB 86,115134 (2012), U. Aschauer et al, PRB 92, 054103 (2015)). Note that we used the Lichtenstein implementation of DFT + U in our work, which yields a $U-J = 4.5 - 1 = 3.5$ eV which is close to the 4 eV value used in the above papers. Given that previous work from the literature indicates that $U \sim 4-5$ eV provides a satisfactory description of structural/magnetic properties of both Mn^{2+} and Mn^{4+} systems, we feel that using the same U value for both is a reasonable choice. The value $J \sim 1$ eV is widely used

to model many transition metal oxides, so using the same J for both Mn oxidation states also is a reasonable choice.

However, to further support this choice, we have performed additional calculations to check the dependence on U . Figure 1 shows the energy difference between the non-polar $Acaa$ and polar $A2_{1am}$ phases of $\text{Ca}_3\text{Mn}_2\text{O}_7$ as a function of U . We find that $Acaa$ is always higher energy, with the energy difference between the two phases varying by about 10 meV/f.u. as we increase U between 3 and 6 eV. Similarly, the magnetic moment on the Mn^{4+} cations stays about the same (~ 2.6 - $2.8 \mu_B$) as we vary U .

Figure 1: Energy difference between $Acaa$ and $A2_{1am}$ phases as a function of U .

We also performed additional calculations on $\text{Ca}_{2.75}\text{Mn}_{2.25}\text{O}_7$ (structure shown in Fig. 5c of the main text) where we changed the value of U on the Mn^{2+} dopant atom while keeping $U = 4.5$ eV on the Mn^{4+} atoms. Due to the presence of partially filled e_g electrons on the Mn^{2+} , we expect that a larger U is needed for this charge state compared to Mn^{4+} . Increasing U on Mn^{2+} to 6 eV produced a negligible change in the energy difference $E_{Acaa} - E_{A2_{1am}}$ (decreased it by 5 meV/f.u.) and the magnetic moment was almost unchanged. Thus these calculations support that using $U=4.5$ eV on both Mn^{2+} and Mn^{4+} cations provides a satisfactory description of the relevant properties for our study.

We have added three sentences to the Methods section (starting at page 19, line 294) describing these additional calculations.

“... in agreement with previous work on $\text{Ca}_3\text{Mn}_2\text{O}_7$.² We have checked that our results are robust against reasonable variations of the U parameter. In addition, we note that in principle describing the A-site Mn^{2+} dopant atoms may require a different value of U compared to the Mn^{4+} atoms. We have checked that our results are robust against varying the U on the Mn^{2+} dopant atoms (while keeping $U = 4.5$ eV on the Mn^{4+} atoms).”

3) The Mn^{2+} in the rock-salt layers are placed in a ferromagnetic order. Again, is it the ground state of the system? Is antiferromagnetism or ferromagnetism does not change the qualitative picture reported in the manuscript (again, as previous comment, I would say not in view of the energies at play that are larger than the magnetic ones but it should be shown to be indeed the case)?

Response: We agree with the referee that the Mn^{2+} cations in the rocksalt layer could have either a ferromagnetic (FM) or antiferromagnetic (AFM) alignment. In addition, the referee is correct to note that the choice of magnetic ordering should have a minimal impact on these energy differences between structural phases that we focus on in this work, since the structural energy scale is much larger than the magnetic one. To verify this point we performed an additional calculation for $\text{Ca}_{2.5}\text{Mn}_{2.5}\text{O}_7$ (shown in Fig. 5d of the main text) where we imposed an antiferromagnetic (AFM) ordering on the Mn^{2+} dopant atoms. We find that the total energy of the $A2_1am$ phase is about 10 meV/f.u. lower when the Mn^{2+} atoms are AFM-ordered compared to FM-ordered, so at least for this particular dopant concentration, AFM ordering is the ground state. We find that the magnetic moment is the same for both magnetic orderings, and the energy difference $E_{Acaa} - E_{A2_1am}$ changes by a small amount (13 meV/f.u.). Thus the choice of Mn^{2+} magnetic ordering does not produce a qualitative difference in our results. Given this, we believe that choosing a FM coupling of the Mn^{2+} spins is the best choice for our calculations because it allows us to compare all dopant concentrations and configurations with the same magnetic ordering imposed (even though it is not always the ground state magnetic order). Using an AFM ordering on the Mn^{2+} spins would introduce the additional complication that some dopant concentrations and configurations would have multiple distinct AFM orderings that must be considered. We have added a sentence to the Methods section (Page 19, starting at line 302) describing this additional calculation.

“We have checked that selecting an antiferromagnetic order for the Mn dopant spins does not produce a qualitative change to our results.”

REVIEWERS' COMMENTS

Reviewer #2 (Remarks to the Author):

The authors have addressed my comments and questions. I think it is ready to publish.

Reviewer #3 (Remarks to the Author):

The authors have satisfyingly replied to all of my comments. They have demonstrated that the qualitative conclusions are stable with the U/J parameter of DFT+U as well as when Mn^{2+} and Mn^{3+} are present in the crystal. They have checked that the magnetic ordering does not affect the qualitative results too. These two main points were addressed carefully and the paper has been updated accordingly such that I do not have any more comment with the new manuscript. Hence, I'm in favor of its publication as it is in Nature communication.